# Natural antisense transcripts of *MIR398* genes suppress microR398 processing and attenuate plant thermotolerance

Yajie Li [1,2], Xiaorong Li[1,2], Jun Yang [1✉] & Yuke He [1✉]

MicroRNAs (miRNAs) and natural antisense transcripts (NATs) control many biological processes and have been broadly applied for genetic manipulation of eukaryotic gene expression. Still unclear, however, are whether and how NATs regulate miRNA production. Here, we report that the *cis*-NATs of *MIR398* genes repress the processing of their pri-miRNAs. Through genome-wide analysis of RNA sequencing data, we identify *cis*-NATs of *MIRNA* genes in Arabidopsis and Brassica. In Arabidopsis, *MIR398b* and *MIR398c* are coexpressed in vascular tissues with their antisense genes *NAT398b* and *NAT398c*, respectively. Knock down of *NAT398b* and *NAT398c* promotes miR398 processing, resulting in stronger plant thermotolerance owing to silencing of miR398-targeted genes; in contrast, their overexpression activates *NAT398b* and *NAT398c*, causing poorer thermotolerance due to the upregulation of miR398-targeted genes. Unexpectedly, overexpression of *MIR398b* and *MIR398c* activates *NAT398b* and *NAT398c*. Taken together, these results suggest that *NAT398b/c* repress miR398 biogenesis and attenuate plant thermotolerance via a regulatory loop.

[1] National Key Laboratory of Plant Molecular Genetics, CAS Center for Excellence in Molecular Plant Sciences, Shanghai Institute of Plant Physiology and Ecology, Chinese Academy of Sciences, 200032 Shanghai, China. [2] University of the Chinese Academy of Sciences, 100049 Beijing, China. ✉email: jyang@cemps.ac.cn; heyk@cemps.ac.cn

MiRNAs (microRNAs) are small RNAs that repress the expression of their target genes at post-transcriptional levels[1]. In plants, appropriate miRNA accumulation depends on the activity of the nuclear proteins HYPONASTIC LEAVES1 (HYL1), DICER-LIKE1 (DCL1), SERRATE (SE), and AGONAUTE1 (AGO1)[2–4]. These proteins are thought to function together to catalyze the processing of pri-miRNAs and pre-miRNAs[5–7]. Nearly all nonlethal mutants of DCL1, HYL1, and SE display multiple morphological, physiological, and biochemical aberrations[8]. Theoretically, the pleiotropic effects of DCL1, HYL1, and SE mutations are attributable to the reduced accumulation of miRNAs and increased expression of miRNA-directed targets[8,9]. Because miRNAs can silence their target genes[10], MIRNA genes and artificial miRNAs have been broadly used for identification of gene function in plants and for genetic manipulation of agricultural traits for high yield and quality[11,12]. For unknown reasons, however, these MIRNA genes frequently do not work in many plants[13].

Natural antisense transcripts (NATs), a class of RNAs containing sequences complementary to sense mRNAs, comprise a set of prominent and complex regulatory RNAs. Recent genomic studies using computational prediction methods and experimental identification approaches have revealed a large number of NATs[14–17]. NATs can be divided into cis-NATs, which are transcribed from opposing DNA strands at the same genomic locus, and trans-NATs, which are transcribed from separate genomic loci. Cis-NATs in turn can be categorized into three types: convergent (3′-end overlap), divergent (5′-end overlap), and enclosed (one transcript encompassing the other)[18,19]. Studies have revealed that cis-NATs can participate in a broad range of regulatory events[20]. Evidence has been uncovered for the involvement of cis-NATs in translation initiation[21,22], mRNA stability[23], transcription termination[24], alternative splicing[25,26], RNA editing[27,28], DNA methylation[29,30], histone methylation[31,32], small interfering RNA (siRNA)-induced gene silencing[33,34], and translational enhancement[35,36]. Luo et al. (2009) have shown that antisense transcription is associated with miRNA-targeted mRNAs in Arabidopsis[37]. Even though several cis-NATs are known to take part in gene regulatory events, the role of many cis-NATs is unclear.

Most documented cis-NATs correspond to protein-coding genes, with only a few cis-NATs of noncoding MIRNA genes having been reported[36,38,39]. Our previous study first identified a subset of cis-NATs corresponding to MIRNA genes in Brassica rapa[16], including BrpMIR398b and BrpMIR398c. miR398, a conserved miRNA that targets CSD1, CSD2, and CCS[40], is proposed to be directly linked to cell death[41] and responses to oxidative stress[13], heat stress[42], water deficit[43], abscisic acid stress[44], high sucrose[45], and bacterial infection[34]. Heat stress can injure a broad spectrum of cellular components and adversely affects the distribution and productivity of horticulturally and agriculturally important plants worldwide[42,46]. Heat stress rapidly induces miR398 and reduces transcripts of its target genes CSD1, CSD2, and CCS. Transgenic plants expressing CSD1, CSD2, and CCS are more sensitive to heat stress than wild-type plants. In contrast, csd1, csd2, and ccs mutant plants are more heat tolerant than the wild-type[42].

In the study reported here, we endeavored to understand the molecular relationship between MIRNA genes and their cis-NATs in miRNA processing and gene regulation in response to high temperature. Functional analysis indicated that the processing of pri-miR398b/c is suppressed by their cis-NATs. This finding provides insight into the function of cis-NATs in miRNA-guided gene silencing and should thus facilitate the genetic manipulation of gene expression for thermotolerance in eukaryotes.

## Results

**Identification of cis-NATs at MIR398 gene loci of B. rapa and Arabidopsis thaliana.** In our previous study, we examined the high-temperature response of 1031 B. rapa cis-NATs, eight of which corresponded to precursors of miRNAs[16]. To explore the genome-wide heat stress response of additional novel cis-NATs of MIRNA genes in B. rapa, we carried out DNA resequencing and RNA sequencing (RNA-seq) of B. rapa ssp. pekinensis 'Bre' (a heading Chinese cabbage) exposed to 42 °C for 0 and 1 h. In combination with our RNA-seq data and B. rapa mRNA sequences deposited in the Brassica Database (http://brassicadb.org/brad/), we identified 22 cis-NATs that were reverse-complementary to MIRNA genes. These cis-NATs were thus designated as cis-NATs of MIRNA genes in B. rapa (Supplementary Table 1).

Four copies of the MIR398 gene—BrpMIR398a-1, BrpMIR398a-2, BrpMIR398b-1, and BrpMIR398b-2—were identified in the Brassica genome (Fig. 1a). Among them, the miRNA precursors of BrpMIR398b-1 and BrpMIR398b-2 were sense genes that overlapped with cis-NATs of BrpNAT398b-1 (Bra006261) and BrpNAT398b-2 (Bra008752), respectively. Analysis of the RNA-seq data revealed that the transcripts of pri-miR398b-2 and BrpNAT398b-2 nearly fully overlapped with each another (Fig. 1a). According to the latest annotations in the Brassica Database, the BrpNAT398b-1 gene encodes a 963-amino-acid protein of a putative polyribonucleotide nucleotidyltransferase. The BrpNAT398b-2 gene encodes a 366-amino-acid protein of unknown function belonging to the Core-2/I-branching beta-1,6-N-acetylglucosaminyltransferase family.

Because A. thaliana is a close relative of B. rapa, we searched for MIRNA genes and their cis-NATs in Arabidopsis using the same criteria used for B. rapa. Twenty-five cis-NATs homologous to B. rapa cis-NATs or reverse-complementary to Arabidopsis MIRNA genes were identified and regarded as cis-NATs of MIRNA genes in A. thaliana (Supplementary Table 1). The Arabidopsis genome contains three copies of the MIR398 gene: MIR398a (AT2G03445), MIR398b (AT5G14545) and MIR398c (AT5G14565) (Fig. 1b). We examined cis-NATs of MIR398b and MIR398c (NAT398b and NAT398c) but failed to identify the cis-NAT of MIR398a. NAT398b is predicted to encode a member of the Core-2/I-branching beta-1,6-N-acetylglucosaminyltransferase family and is homologous to BrpNAT398b-2 in B. rapa, with the two genes sharing 74.3% and 84.4% nucleic acid and amino acid sequence identities, respectively. A proteome analysis revealed that NAT398c encodes high-affinity nitrate transporter 2.7[47]. To obtain full-length transcripts of MIR398b and MIR398c genes and their cis-NATs, we performed a RACE assay and found that the longest transcripts of pri-miR398b, pri-miR398c, NAT398b, and NAT398c were 692, 1941, 1772, and 1917 nt, respectively (Fig. 1b). In addition, we noticed that pri-miR398b nearly full overlapped with NAT398b (Supplementary Fig. 2a), while pri-miR398c partially overlapped with NAT398c (Supplementary Fig. 2b).

We further discovered that MIR398 genes and their cis-NATs are conserved in angiosperms, especially Brassicaceae. All 12 analyzed species in this plant family were found to have MIR398 genes and their cis-NATs. Most of these NAT398 genes encode core-2/I-branching enzyme, nitrate transporter 2.7 or putative polyribonucleotide nucleotidyltransferase, as in B. rapa and A. thaliana (Supplementary Data 2). These results imply that MIR398 and NAT398 genes are highly conserved in cruciferous plants.

**Overexpression of MIR398b and MIR398c genes fails to elevate miR398 levels.** The MIR398 gene family in Arabidopsis comprises MIR398a, MIR398b, and MIR398c, but MIR398a is not

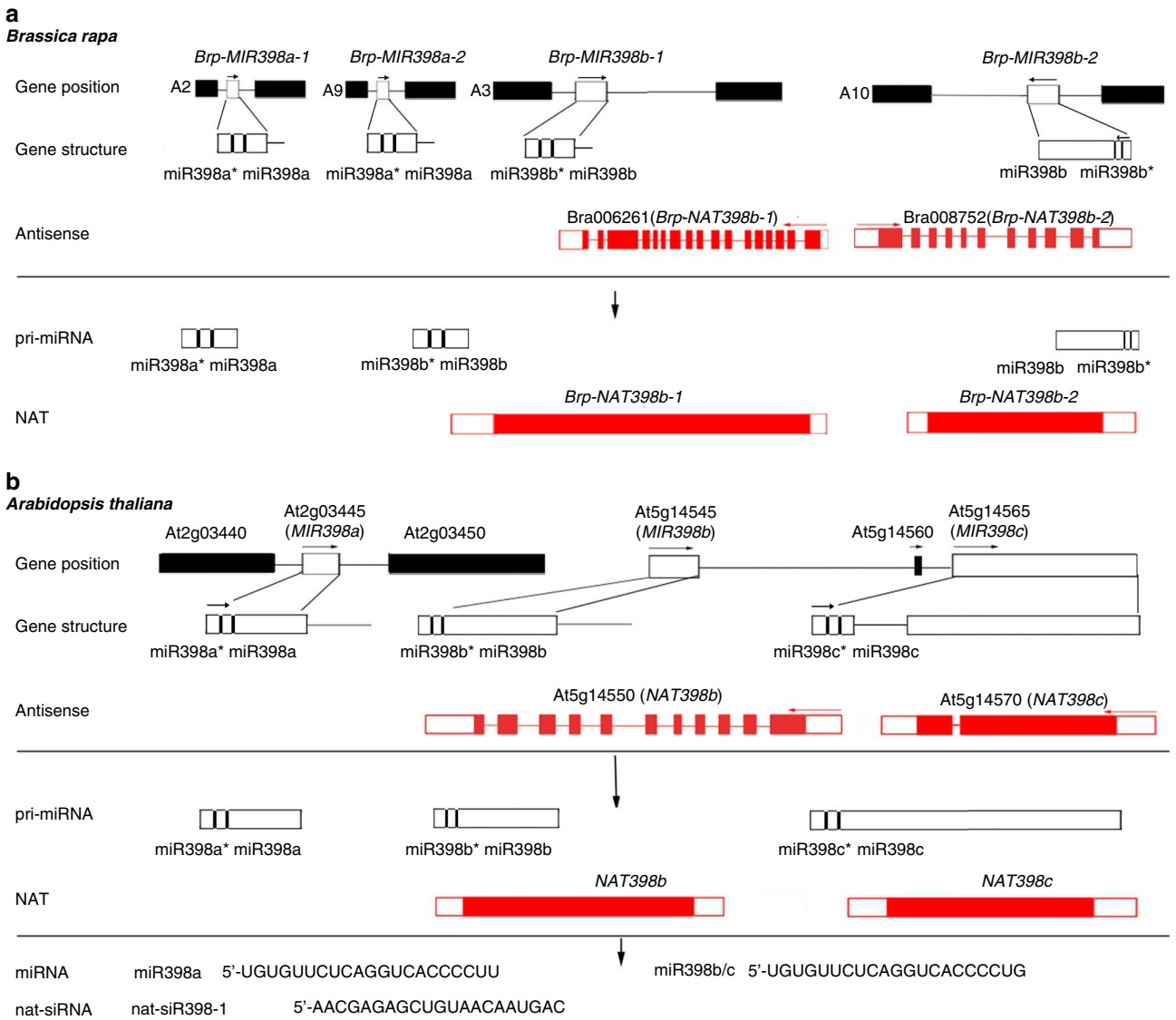

**Fig. 1 Schematic positions and structures of *MIR398* genes and their cis-NATs in *Arabidopsis thaliana* and *Brassica rapa*. a, b** Schematic positions and structures of *BrpMIR398* genes and their *cis*-NATs in *B. rapa* (**a**) and *MIR398* genes and their *cis*-NATs in *A. thaliana* (**b**). Flanking genes, *MIRNA* genes and exons are enclosed in black, white and red boxes, respectively. Solid lines indicate introns and intergenic regions. Arrows indicate transcription directions.

expressed[48]. As revealed by our real-time quantitative reverse transcription PCR (qRT-PCR) experiment and sRNA-seq, the expression level of *MIR398a* was extremely low (Supplementary Fig. 3 and Supplementary Table 2), while *MIR398b* and *MIR398c* were very strongly expressed under the same experimental conditions.

To explore the temporal and spatial expression patterns of *MIR398* and *NAT398*, the promoters of *MIR398a/b/c* and *NAT398b/c* fused with the GUS gene were transferred separately into the Col ecotype of Arabidopsis. In T₃ seedlings, *pMIR398a::GUS* expression was examined in hypocotyls and leaf tips (Supplementary Fig. 3b). Both *pMIR398b::GUS* and *pNAT398b::GUS* were constitutively expressed, with preferential expression observed in vascular tissues (Fig. 2a). *pMIR398c::GUS* expression was detected in all tissues, especially vascular tissues of seedlings (Fig. 2b), whereas *pNAT398c::GUS* expression was restricted to the vascular tissues of cotyledons and hypocotyls (Fig. 2b). These results indicate that *pMIR398b::GUS* and *pNAT398b::GUS* were co-expressed in plant tissues, while *pMIR398c::GUS* and *pNAT398c::GUS* were partially co-expressed in cotyledons and hypocotyls. The similar expression patterns between

*pMIR398::GUS* and *pNAT398::GUS* suggest the possibility of shared regulatory mechanisms.

We separately overexpressed *MIR398a*, *MIR398b*, and *MIR398c* genes in Col under the control of the CaMV 35 S promoter. Overexpression of *MIR398a* was associated with higher levels of miR398 and lower levels of the miR398-targeted *CSD1* gene compared with the wild-type (Fig. 2g, h and Supplementary Table 3). In repeated trials, miR398 levels unexpectedly declined in transgenic lines of *p35S::MIR398b* and *p35S::MIR398c*, causing upregulation of the *CSD1* gene (Figs. 2c–f and S4a–b). Because *MIR398b* and *MIR398c* primary transcript levels were higher in *p35S::MIR398b* and *p35S::MIR398c* lines compared with wild-type plants, they were obviously not silenced in these transgenic lines. We thus wondered whether the unexpected decline in miR398 levels in *p35S::MIR398b* and *p35S::MIR398c* transgenic lines was due to the efficiency of *p35S::MIR398b* and *p35S::MIR398c* constructs, which may not have been able to properly produce miR398. We therefore transiently expressed *p35S::MIR398a*, *p35S::MIR398b*, and *p35S::MIR398c* constructs in tobacco (*Nicotiana benthamiana*). In the transgenic tobacco plants, the accumulation of miR398a and miR398b/c was higher

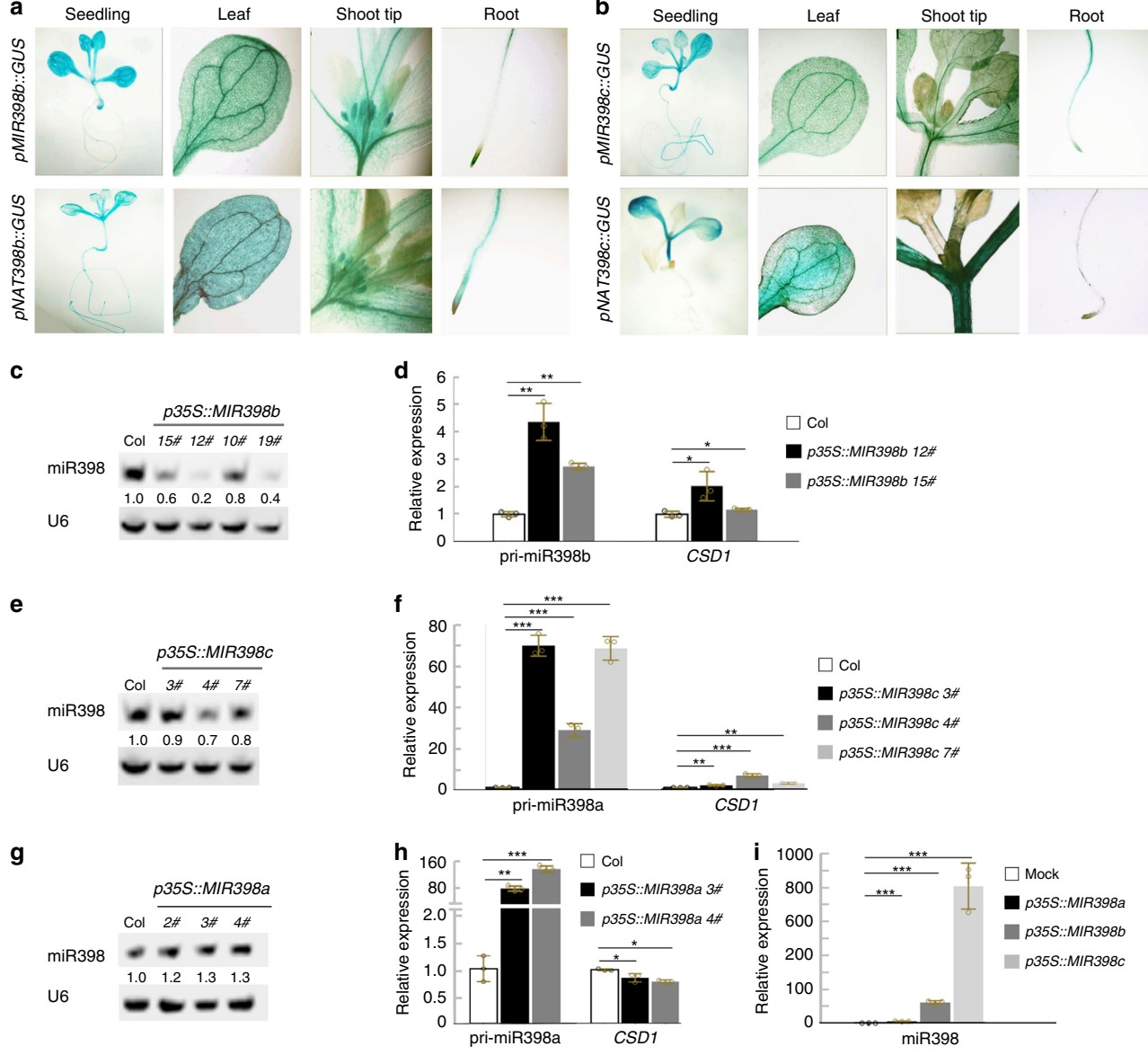

**Fig. 2 Reduction in miR398 levels by overexpression of *MIR398b* and *MIR398c* genes. a, b** GUS signals showing temporal and spatial expression of *MIR398b* and *NAT398b* (**a**) and *MIR398c* and *NAT398c* (**b**). The experiments were repeated at least three times with similar results, and data from one representative experiment are shown. **c, e, g** Northern blot showing miR398 accumulation in *p35S::MIR398b* (**c**), *p35S::MIR398c* (**e**), and *p35S::MIR398a* (**g**) seedlings. **d, f, h** Results of qRT-PCR showing relative expression levels of related genes in *p35S::MIR398b* (**d**), *p35S::MIR398c* (**f**), and *p35S::MIR398a* (**h**) seedlings. **i** miR398 accumulation in tobacco plants transiently expressing *MIR398* genes. Error bars are the mean ± SD. Significant differences were determined by one-tailed student's *t*-test (\**P* < 0.05; \*\**P* < 0.01; \*\*\**P* < 0.001).

than in control plants (Fig. 2i), thus indicating that all of the *MIR398* overexpression constructs were functioning in plants. We therefore inferred that some genetic elements in *A. thaliana* may be involved in repressing miR398 processing in *p35S:: MIR398b* and *p35S::MIR398c* plants but not in *p35S::MIR398a* plants.

**Knock down of *NAT398b/c* upregulates *MIR398b* and *MIR398c*.** To test whether *NAT398b* and *NAT398c* are involved in regulating the expression of *MIR398b* and *MIR398c*, we knocked down the expression levels of *NAT398b* and *NAT398c* by constructing artificial miRNAs of *NAT398b* and *NAT398c*. *NAT398b* overlaps with *MIR398b* in pre-miR398b regions, while *NAT398c* overlaps with *MIR398c* elsewhere. To design artificial miRNAs, we selected appropriate fragments in upstream regions of *NAT398b* and *NAT398c*. *p35S::amiR-NAT398b*, an artificial miRNA specific for

*NAT398b* in a *MIR319a* backbone[12], yielded three transgenic lines when transferred into the Col ecotype. Expression levels of *NAT398b* declined in *p35S::amiR-NAT398b* plants concomitant with an increase in pri-miR398b (Fig. 3a). Decreased expression of *NAT398c* and increased levels of pri-miR398c were also observed in transgenic lines of *p35S::amiR-NAT398c* plants (Fig. 3b). In addition, relative expression levels of *CSD1* decreased in *p35S::amiR-NAT398b* and *p35S::amiR-NAT398c* plants (Fig. 3a–b). These results revealed that *NAT398b* and *NAT398c* negatively regulate pri-miR398b and pri-miR398c levels.

To avoid the possibility of bias, we used the CRISPR/Cas9 approach to edit *NAT398b*. The resulting transgenic lines were designated as *CRNAT398b* plants. The proteins of the Core-2/I-branching enzyme encoded by *NAT398b* were truncated because of premature translational termination, but this had no effect on the expression level of pri-miR398b (Figs. 3e and S5). This

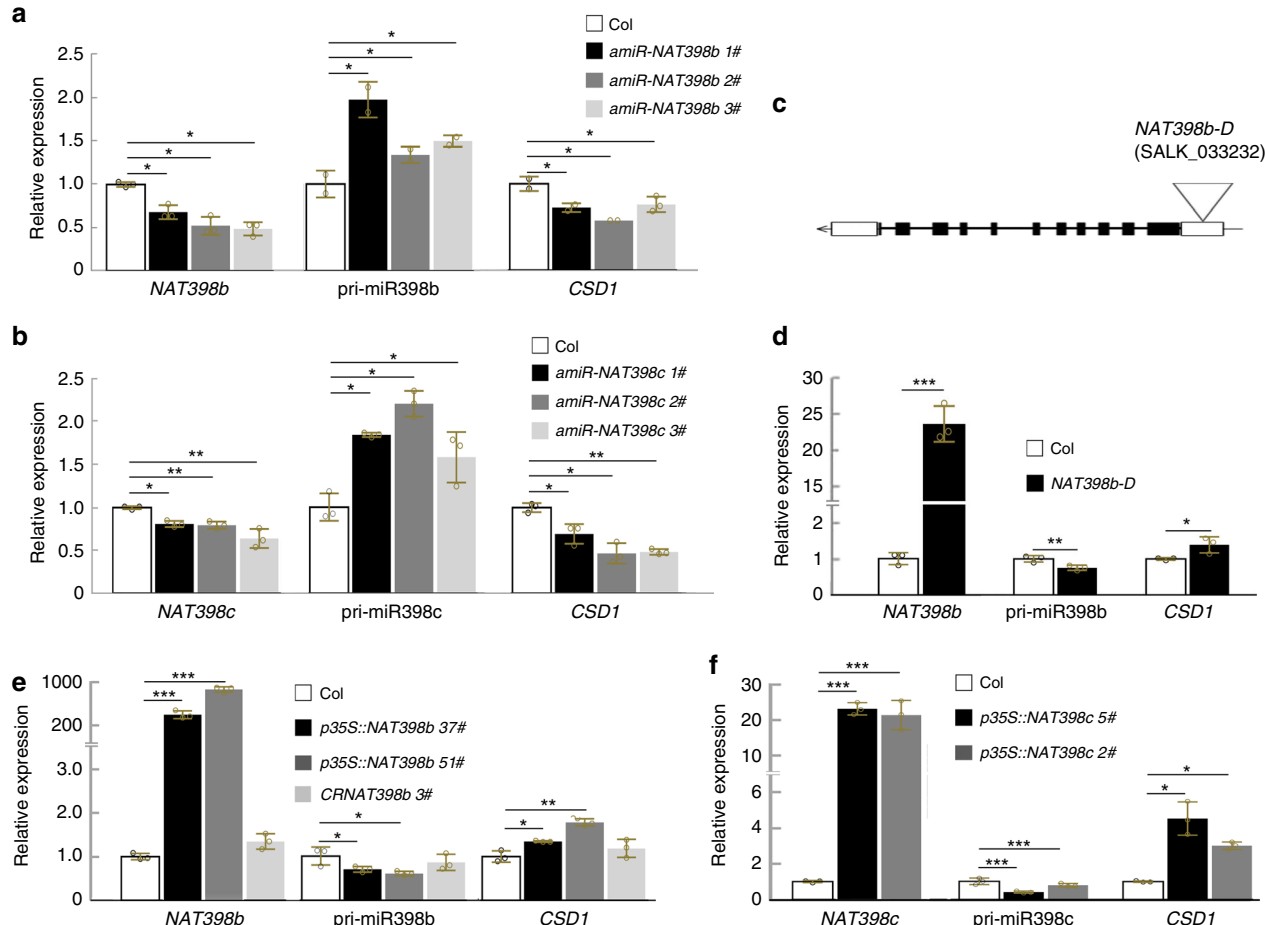

**Fig. 3 Reduction in pri-miR398b/c levels by *NAT398b/c*. a, b** Relative expression levels of related genes in *p35S::amiR-NAT398b* (**a**) and *p35S::amiR-NAT398c* (**b**) seedlings. **c** Diagram showing the location of the T-DNA insertion in the *NAT398b-D* mutant. **d** Relative expression levels of related genes in *NAT398b-D* seedlings. **e, f** Relative expression levels of genes in *p35S::NAT398b* and *CRNAT398b* (**e**) and *p35S::NAT398c* (**f**) seedlings. Error bars are the mean ± SD. Significant differences were determined by one-tailed student's *t*-test (*$P < 0.05$; **$P < 0.01$; ***$P < 0.001$).

result excluded the possibility that pri-miR398b was affected by NAT398b proteins.

We crossed the *p35S::MIR398b* plants with *p35S::amiR-NAT398b* or *CRNAT398b* plants to analyze the genetic interaction between *MIR398b* and *NAT398b*. Expression levels of pri-miR398b and miR398 in *p35S::MIR398b × p35S::amiR-NAT398b* plants were significantly higher than those in *p35S::MIR398b* plants (Supplementary Fig. 6), whereas levels in *p35S::MIR398b × CRNT398b* plants were not significantly higher than those in *p35S::MIR398b* plants (Supplementary Fig. 6). These results indicate that NAT398b transcripts, rather than NAT398b proteins, impaired the expression of pri-miR398b and miR398 in *p35S::MIR398b* plants.

**Overexpressing *NAT398b* and *NAT398c* represses the processing of miR398**. To test how *NAT398b* and *NAT398c* regulate pri-miR398b and pri-miR398c, we next overexpressed *NAT398b* and *NAT398c* in Col. In *p35S::NAT398b* plants, the upregulation of *NAT398b* was concomitant with downregulation of pri-miR398b (Fig. 3e), which caused a decline in miR398 accumulation (Supplementary Fig. 7a and Supplementary Table 3) and upregulated *CSD1* expression (Figs. 3e and S4c); this indicates that *NAT398b* negatively regulates pri-miR398b. We additionally selected *NAT398b-D*, a dominant mutant of *NAT398b* (*NAT398b-D*, SALK_033232) (Fig. 3c). In *NAT398b-D* plants, the expression of *NAT398b* increased, while pri-miR398b declined

and *CSD1* was upregulated (Fig. 3d). Similarly, overexpression of *NAT398c* had negative effects on pri-miR398c (Figs. 3f, S4d, and S7b).

**NAT398b/c impair the stability and cleavage processing of pri-miR398b/c**. To ascertain whether *MIR398b* and *MIR398c* genes are transcriptionally regulated by their *cis*-NATs, we investigated the occupancy of RNA polymerase II (Pol II) at the promoters of *MIR398b* and *MIR398c* in *p35S::NAT398b* and *p35S::NAT398c* plants, respectively. No significant difference was observed in Pol II occupancy between wild-type and transgenic plants (Supplementary Fig. 8). This result suggests that *NAT398b* and *NAT398c* do not regulate *MIR398b* and *MIR398c* genes at the transcriptional level.

The regulation of *MIRNAs* by their *cis*-NATs might also take place via the formation of an RNA duplex, which alters the secondary or tertiary structure of RNA[23,49,50]. To determine whether NAT398b/c and pri-miR398b/c form dsRNA, we devised an in vivo RNase protection assay based on the differential susceptibility of single-stranded RNAs (ssRNAs) and double-stranded RNAs (dsRNAs) to RNase A + T[51]. RNase-treated or untreated Col RNA was used as a template for strand-specific qRT-PCR along with primers located in complementary or non-complementary regions. According to the results, non-complementary regions of NAT398b/c and pri-miR398b/c were degraded by RNase A + T, while complementary regions were

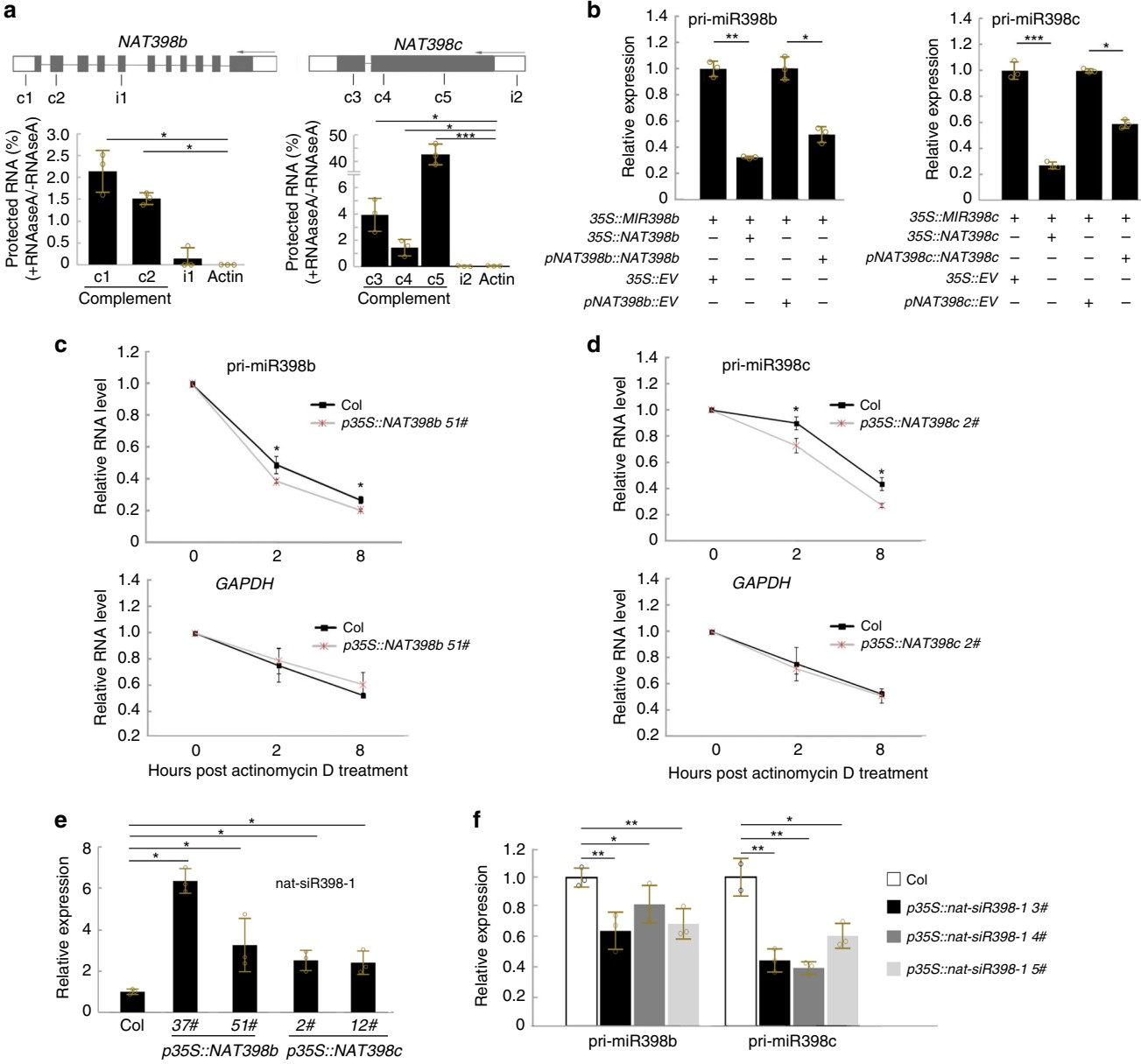

**Fig. 4 Mechanisms of regulation of pri-miR398b/c by *NAT398b/c*. a** RNase protection assays showing double-stranded RNA formation between *NAT398b/c* and pri-miR398b/c. Primers c1, c2, c3, c4, and c5 are located in complementary regions between pri-miRNAs and NAT RNAs; primers i1 and i2 are located in non-complementary regions. **b** Relative expression levels of *MIR398b/c* in tobacco plants transiently expressing *NAT398b/c* and *MIR398b/c*. **c, d** Relative RNA levels at different time points after treatment with actinomycin D (100 μg/mL) in Col and *p35S::NAT398b* (**c**) and *p35S::NAT398c* (**d**) lines. **e** nat-siR398b-1 accumulation in *p35S::NAT398b* and *p35S::NAT398c* seedlings. **f** Relative expression levels of *MIR398b* and *MIR398c* in *p35S::nat-siR398b-1* seedlings. Error bars are the mean ± SD. Significant differences were determined by one-tailed student's *t*-test (*$P < 0.05$; **$P < 0.01$; ***$P < 0.001$).

protected from degradation (Fig. 4a). Similar degradation was observed in the case of transcripts of ACTIN2, used as a negative control. These results suggest that pri-miR398b/c and their *cis*-NATs form double-stranded RNAs in vivo.

Because RNA duplexes have been reported to influence RNA stability[23,49,50], we examined whether NAT398b/c regulate pri-miR398b/c RNA stability. We first transiently expressed *p35S::MIR398b* synchronously with *p35S::NAT398b* or *pNAT398b::NAT398b* in tobacco and found that the expression level of *MIR398b* was downregulated compared with plants expressing *p35S::MIR398b* or empty vectors (Fig. 4b). We then co-expressed *p35S::MIR398b* and *p35S::NAT398b* in Col and found that the expression level of *MIR398b* was also downregulated relative to

plants only expressing *p35S::MIR398b* (Supplementary Fig. 9). A similar inhibitory effect on *MIR398c* due to *NAT398c* was observed when we co-expressed *p35S::MIR398c* and either *p35S::NAT398c* or *pNAT398c::NAT398c* in tobacco and Arabidopsis (Figs. 4b and S9). These results indicate that *NAT398b* and *NAT398c* impair the stability of pri-miR398b and pri-miR398c transcripts, respectively.

We further measured the RNA decay rate of pri-miR398b/c in Col, *p35S::NAT398b* and *p35S::NAT398c* lines via treatment with the transcriptional inhibitor actinomycin D. We found that the decay rates of pri-miR398b and pri-miR398c were higher in *p35S::NAT398b* and *p35S::NAT398c* plants than in Col (Figs. 4c, d). In contrast, the decay rates of GAPDH transcripts in

Col, *p35S::NAT398b* and *p35S::NAT398c* lines were indistinguishable (Fig. 4c, d). These results confirm that *NAT398b* and *NAT398c* impair the stability of pri-miR398b and pri-miR398c transcripts, respectively.

Aberrance of cleavage sites in pri-miRNAs causes alterations in pre-miRNA sequences, which likely leads to the production of inaccurate miRNAs. To understand whether *NAT398* affects pri-miR398 cleavage, we collected RNA samples of 3-week-old seedlings and carried out a 5′ RACE-PCR analysis to detect the cleavage sites in 5′ single-stranded RNA regions of pri-miR398b. The percentage of inaccurate cleavage sites in *p35S::amiR-NAT398b* plants was lower than in the wild-type, whereas the percentage was much higher in *p35S::NAT398b* lines (Supplementary Table 4). Small RNA (20–70 nt) paired-end sequencing further revealed a subset of degradation products from pri-miR398b, and the accumulation of these fragments was also much higher in *p35S:: NAT398b* plants than in the wild-type (Supplementary Table 5). These results suggest that *NAT398b* overexpression increases the number of incorrect pri-miR398b cleavage sites and thus interferes with its accurate processing.

The 3′-UTR region of *NAT398b* is complementary to pri-miR398b, but its coding sequence is not. To examine whether the *NAT398b* 3′-UTR region affects the expression of *MIR398b*, we overexpressed *NAT398b* cDNA and *NAT398b* coding sequences in Col. In this way, we generated a version of *NAT398b* that maintains its ORF but loses 315 nt sequence complementarity to pri-miR398b. In *p35S::NAT398b(cDNA)* plants, *NAT398b* was upregulated, concomitant with downregulation of pri-miR398b (Supplementary Fig. 10). In contrast, the upregulation of *NAT398b* in *p35S::NAT398b(CDS)* plants did not accompany downregulation of pri-miR398b (Supplementary Fig. 10). This result suggests that the *NAT398b* 3′-UTR region complementary to pri-miR398b is crucial for pri-miR398b expression.

**Overexpression of nat-siR398 reduces pri-miR398b and pri-miR398c levels.** The regulation of *MIR398b/c* by their *cis*-NATs may be also achieved via the formation of *cis*-NAT-derived small interfering RNAs (nat-siRNAs), which may degrade pri-miR398b/c through siRNA-induced gene silencing. sRNA deep sequencing revealed several types of nat-siR398 entities—the siRNAs derived from *NAT398* transcripts. Among them, nat-siR398b-1 and nat-siR398b-2 are located in the region of *NAT398b* and *MIR398b* overlap (Fig. 1b). Both nat-siRNAs were accumulated at much higher levels in *p35S::NAT398b* plants than in the wild-type (Figs. 4e and S11b).

To define genetic components required for the formation of nat-siRNAs, the expression level of nat-siR398b-1 was examined in some small RNA biogenesis mutants. We found that *dcl2/3/4* triple mutations or the *rdr6* mutation blocked nat-siR398b-1 accumulation, while *dcl1*, *dcl2*, *dcl3*, *dcl4*, and *rdr2* single mutations had little effect on nat-siR398b-1 accumulation (Supplementary Fig. 12). These results suggest that RDR6 is required for nat-siR398b-1 accumulation and that DCL2, DCL3 and DCL4 play redundant roles in nat-siR398b-1 production.

To determine the function of nat-siR398b-1, we constructed an artificial miRNA vector with a *MIR319a* backbone[12] and overexpressed nat-siR398b-1 in Col under the control of the CaMV 35S promoter. Expression levels of both pri-miR398b and pri-miR398c declined in the *p35S::nat-siR398b-1* plants (Fig. 4f). This fact suggests that nat-siR398-1 reduces pri-miR398b and pri-miR398c levels.

**Expression of NAT398 is upregulated by overexpression of MIR398 genes.** To examine whether the expression of *MIR398b/c* genes affects the expression of their antisense genes, we investigated *NAT398b/c* expression in *p35S::MIR398b/c* plants. According to our data, expression levels of *NAT398b* and *NAT398c* were much higher in *p35S::MIR398b* and *p35S::MIR398c* plants than in the wild-type (Fig. 5a, b). We also discovered that expression levels of *NAT398b* and *NAT398c* were reduced in *p35S::nat-siR398-1* plants (Fig. 5c). These results indicate that *MIR398b* and *MIR398c* genes positively regulate their antisense genes.

SALK_038698 mutants harboring a T-DNA insertion in the promoter region of *MIR398c* were applied to confirm the above finding (Fig. 5d). The SALK_038698 mutant displayed lower *MIR398c* expression and was designated as *mir398c* (Figs. 5e and S7c). The expression of *NAT398c* was downregulated in the *mir398c* mutant (Fig. 5e). These results confirm that the expression of *MIR398b* and *MIR398c* genes positively regulates their respective *cis*-NATs.

We next examined how *MIR398b* and *MIR398c* regulate their *cis*-NATs. We found that *MIR398b* and *MIR398c* overexpression enhanced the transcriptional activity of *NAT398b* and *NAT398c*, respectively (Fig. 5a, b). ChIP-qPCR analysis also revealed that recruitment of Pol II at the promoter of *NAT398b* or *NAT398c* was respectively increased in *p35S::MIR398b* and *p35S::MIR398c* plants (Fig. 5g). These results suggest that *MIR398b* and *MIR398c* directly activate the transcription of their *cis*-NATs.

To reveal differences in the RNA interference of NATs between pri-miR398 and mature miR398, we constructed an artificial miR398b/c (*amiR398b/c*) vector with a *MIR319a* backbone[12]. In *p35S::amiR398b/c* seedlings, *CSD1* and *CSD2* expressions were notably downregulated compared with the wild-type (Fig. 5f), whereas *NAT398b* and *NAT398c* were unchanged (Fig. 5f). This fact implies that mature miR398 can silence its target genes instead of its *cis*-NATs.

**NAT398b/c attenuate plant thermotolerance.** Previous researchers have reported that *cis*-NATs regulate plant temperature response[52,53]. In addition, pri-miR398b, pri-miR398c and mature miR398 are induced by heat stress in Arabidopsis[42]. To examine whether the *cis*-NATs of *MIR398b/c* are heat-responsive, we subjected Col seedlings to a heat treatment of 38 °C for 1 h. Northern blotting revealed that miR398 accumulation in seedlings was much higher at 38 °C than at 22 °C (Fig. 6a). Real-time PCR indicated that pri-miR398b and pri-miR398c expressions were induced by heat stress, while *NAT398b* and *NAT398c* expressions were inhibited (Fig. 6b). At the same time, the expression of the miR398-targeted *CSD1* gene was downregulated under heat treatment (Supplementary Fig. 13c). Under heat treatment, expression levels of pri-miR398b and miR398 were lower and those of *CSD1* were higher in *p35S::NAT398b* plants than in the wild-type (Supplementary Fig. 13a–c), thus indicating that *NAT398b* affects heat responses of miR398 and *CSD1*.

To gain insight into the mechanism of *MIR398/NAT398* regulation under heat stress, *MIR398b/c*- and *NAT398b/c*-promoter–GUS transgenic plants were subjected to the same heat stress conditions and analyzed for GUS activity. Analysis of the seedlings revealed an increase in GUS signal intensity after stress treatment in *pMIR398b/c::GUS* but a decrease in *pNAT398b/c:: GUS* (Supplementary Fig. 13d). These results indicate that heat stress activates the transcription of *MIR398b/c* genes and suppresses the transcription of *NAT398b/c* genes.

The redox status of plants is reported to be influenced by heat stress, which affects expression of heat-responsive genes[42,54–56]. We thus measured $H_2O_2$ levels in mutant and transgenic plants. Similar to the findings of Guan et al. (2013), *csd1* plants accumulated much higher levels of $H_2O_2$ (as indicated by 3,3′-diaminobenzidine staining) under heat stress (Fig. 6c), while *p35S::CSD1* plants accumulated much lower $H_2O_2$ levels (Supplementary Fig. 15).

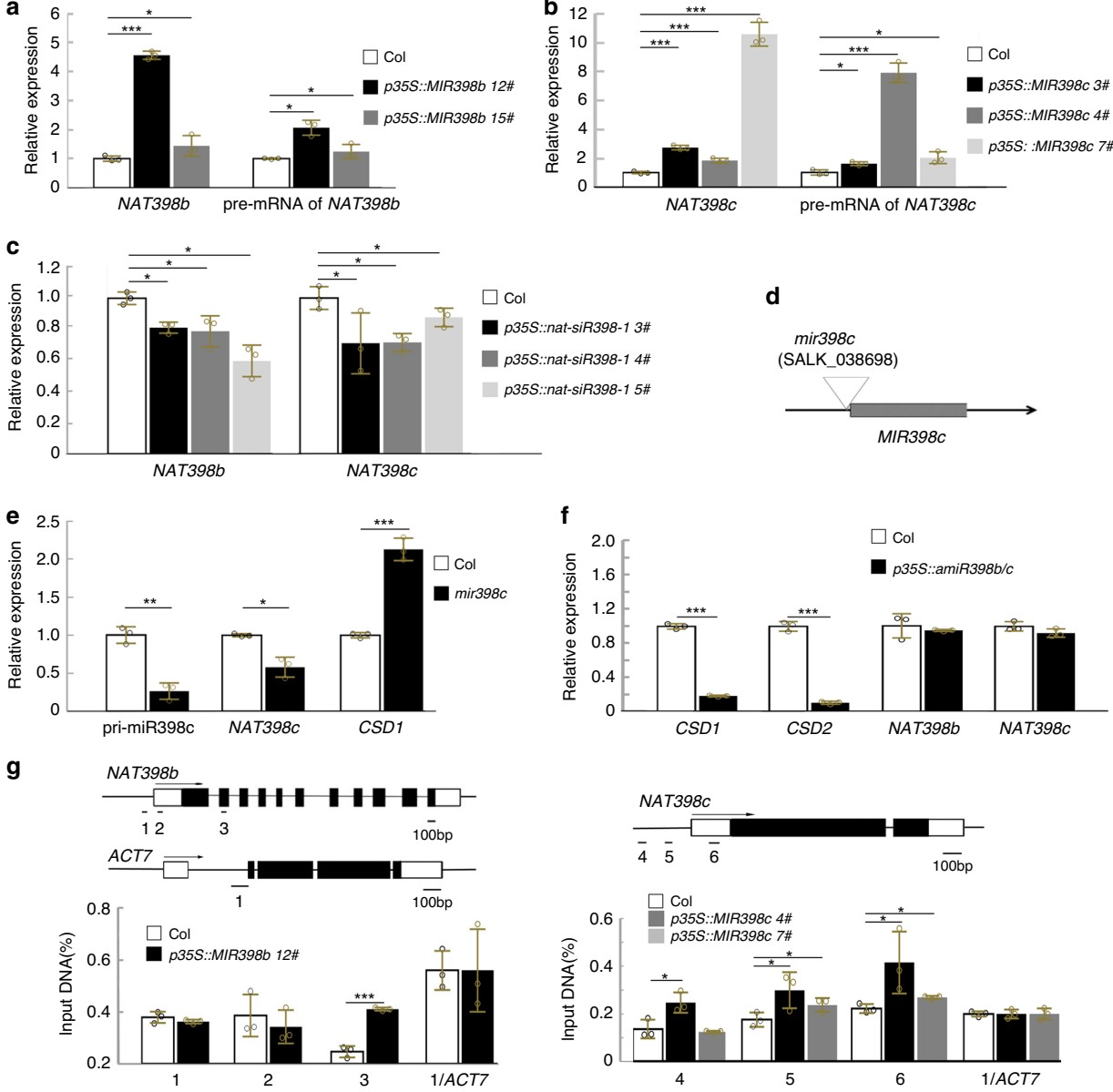

**Fig. 5 MIR398b and MIR398c activation of their cis-NATs. a, b** Relative expression levels of genes in *p35S::MIR398b* (**a**) and *p35S::MIR398c* (**b**) seedlings. **c** Relative expression levels of *NAT398b* and *NAT398c* in *p35S::nat-siR398b-1* seedlings. **d** Diagram showing the location of the T-DNA insertion in the *mir398c* mutant. **e, f** Results of qRT-PCR showing gene relative expression levels in *mir398c* mutant (**e**) and *p35S::amiR398b/c* (**f**) seedlings. **g** ChIP analysis showing relative accumulation of RNA Pol II in *p35S::MIR398b* and *p35S::MIR398c* plants. Error bars are the mean ± SD. Significant differences were determined by one-tailed student's *t*-test (*$P < 0.05$; **$P < 0.01$; ***$P < 0.001$).

Compared with the wild-type, *p35S::MIR398a* plants accumulated slightly higher levels of $H_2O_2$ under control conditions and much higher levels under heat stress (Supplementary Fig. 15). In contrast, $H_2O_2$ accumulation was lower in *p35S::MIR398b* and *p35S:: MIR398c* plants under both control and heat stress conditions compared with the wild-type (Supplementary Fig. 15). These results indicate that altered expression of miR398 changed the redox status of plants.

We also investigated whether *NAT398b/c* could alter the redox status of transgenic plants under heat stress. Compared with wild-type plants, *p35S::amiR-NAT398b/c* plants accumulated slightly higher levels of $H_2O_2$ under control conditions (Fig. 6c) and much higher levels under heat stress (Fig. 6c). In contrast, *p35S:: NAT398b* and *p35S::NAT398c* plants accumulated slightly lower levels of $H_2O_2$ under control conditions compared with wild-type

plants and much lower levels under heat stress (Fig. 6c). $H_2O_2$ accumulation in *CRNAT398b 3#* mutants was similar to wild-type plants under both control and heat stress conditions (Fig. 6c). Taken together, these results demonstrate that *NAT398b* and *NAT398c* alter the redox status of plants by regulating miR398 accumulation.

Next, we examined the thermotolerance of mutants and transgenic plants. We found that *csd1* mutants were insensitive to heat stress, as indicated by increased survival rates of flowers and increased percentages of green leaves under heat stress (Fig. 6d–f). In contrast, transgenic plants overexpressing *CSD1* gene displayed increased sensitivity to heat shock (Supplementary Fig. 16). We also measured the thermotolerance of *p35S:: MIR398a*, *p35S::MIR398b* and *p35S::MIR398c* plants. We found that *p35S::MIR398a* plants had stronger heat resistance,

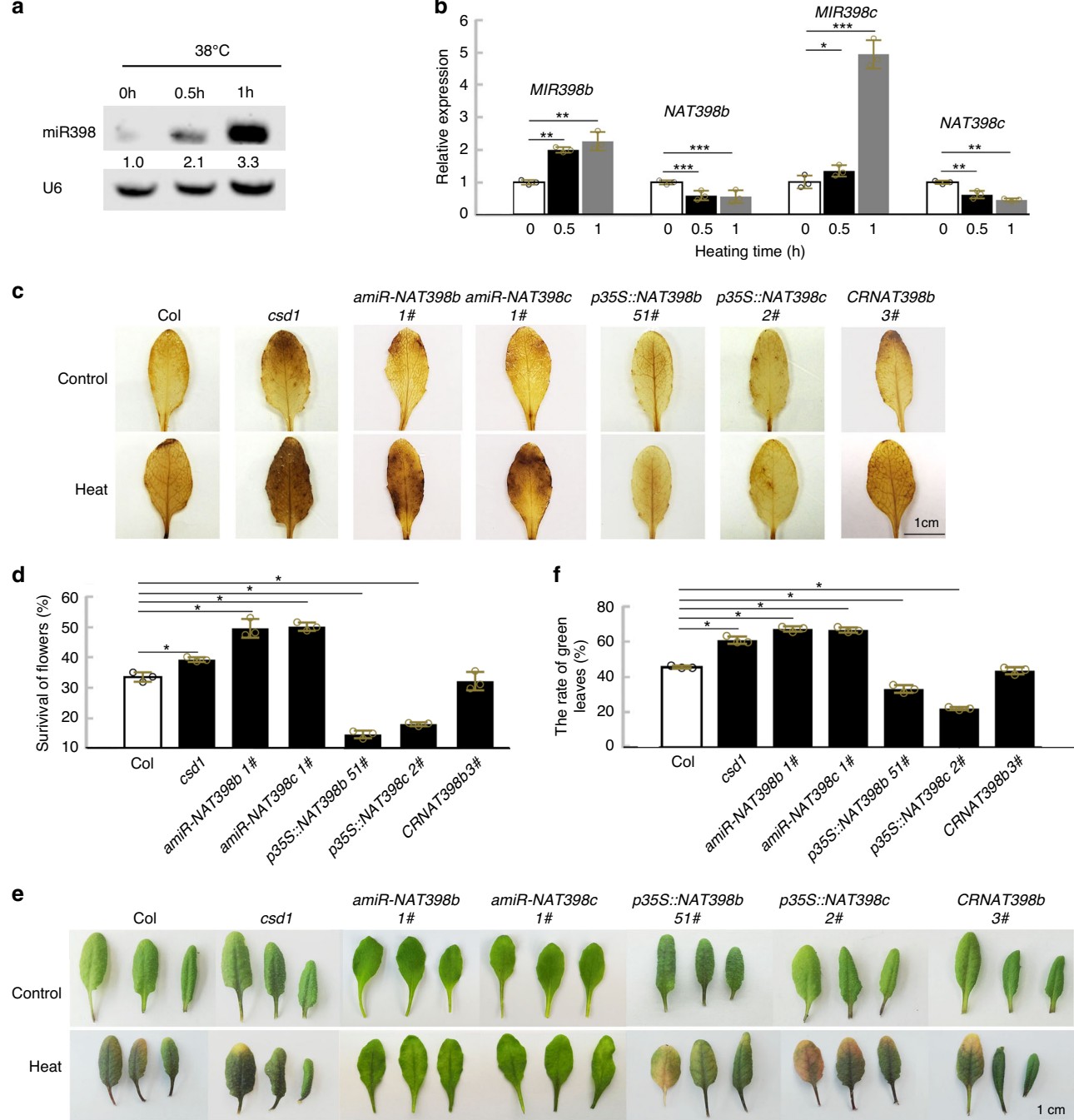

**Fig. 6 NAT398b/c attenuation of H₂O₂ accumulation and thermotolerance in Arabidopsis. a** Northern blot showing miR398 accumulation in seedlings exposed to high temperature for 0 to 1 h. **b** Relative expression levels of *MIR398b*, *NAT398b*, *MIR398c*, and *NAT398c* in seedlings exposed to high temperature for 0–1 h. Error bars are the mean ± SD (*n* = 3 biological replicates). **c** DAB staining for H₂O₂ levels in transgenic and mutant plants. **d** Survival rate of flowers of wild-type, *csd1*, *p35S::amiR-NAT398b*, *p35S::amiR-NAT398c*, *p35S::NAT398b*, *p35S::NAT398c*, and *CRNAT398b* plants under heat stress. **e, f** Phenotypes of detached leaves (**e**) and the rate of green leaves (**f**) of wild-type, *csd1*, *p35S::amiR-NAT398b*, *p35S::amiR-NAT398c*, *p35S::NAT398b*, *p35S::NAT398c*, and *CRNAT398b* plants. Error bars are the mean ± SD (*n* = 3 biological replicates, a replicate constitutes 60 biologically independent samples for **d**, and 12 biologically independent samples for **f**). Significant differences were determined by one-tailed student's *t*-test (*P < 0.05; **P < 0.01; ***P < 0.001).

whereas *p35S::MIR398b* and *p35S::MIR398c* plants were heat sensitive, as expected (Supplementary Fig. 16). These results indicate that the miR398-CSD1 regulon modulates plant thermotolerance.

We additionally measured the thermotolerance of *p35S::amiR-NAT398b*, *p35S::amiR-NAT398c*, *p35S::NAT398b*, *p35S::NAT398c*, and *CRNAT398b* plants. According to our measurements, *p35S::amiR-NAT398b* and *p35S::amiR-NAT398c* plants were more

tolerant to heat stress compared with the wild-type, while *p35S::NAT398b* and *p35S::NAT398c* plants were more sensitive (Fig. 6d–f). *CRNAT398b* plants, however, exhibited no obvious differences compared with wild-type plants (Fig. 6d–f). These results suggest that *NAT398b/c* attenuate plant thermotolerance by regulating pri-miR398 processing and the expression of miR398 target genes, thus indicating the important role of *NAT398b/c* in the fine-tuning of plant thermotolerance.

## Discussion

Gene expression regulated by miRNAs and *cis*-NATs is well known in eukaryotes[1,57], but the molecular relationship between *MIRNA* genes and their *cis*-NATs has rarely been documented[38]. In the present study, *MIR398* genes in both *B. rapa* and *A. thaliana* were found to have *cis*-NATs. Sense–antisense regulatory loops between coding genes and long noncoding genes are usually widely present[58]. Here, we uncovered the existence of regulatory loops between coding sequences and pri-miRNAs.

In our previous study, we identified eight *cis*-NATs responsive to high temperature in *B. rapa* that correspond to precursors of miRNAs[16]. Using a combination of RNA-seq data and mRNA sequences, we identified 22 *cis*-NATs complementary to *MIRNA* genes in *Brassica rapa* and 25 *MIRNA cis*-NATs in Arabidopsis. We further found that *MIR398* genes and their *cis*-NATs are conserved in Brassicaceae species, which indicates that *cis*-NATs of *MIRNA* are widely present in plants. Thousands of NATs have been identified in plants and implicated in seedling light response[59], flowering time control[60], reproduction[61], and biotic and abiotic stress responses[31,33,34,53,62,63], but only a handful have been experimentally characterized. We expect that an increasing number of *cis*-NATs of *MIRNA* will be discovered under various conditions.

MiRNAs are derived from noncoding *MIRNA* genes and control plant growth and physiology through silencing of their target genes[64,65]. For genetic manipulation of agricultural traits, *MIRNA* genes have been broadly utilized[10]. Unfortunately, many *MIRNA* vectors do not work in transgenic plants. For example, overexpression of the *MIR398b* gene in Arabidopsis fails to elevate miR398 accumulation[13]. For identification of *MIRNA* gene functions in plants, artificial miRNAs are designed according to changes in RNA secondary or tertiary structures[12]. How these *MIRNA* genes lose their miRNA biogenesis ability remains unclear. Long noncoding RNAs (lncRNAs) have been reported to inhibit the expression of mature-Sly-miR482a in *Solanum lycopersicum*[39]. In the present study, we found that miRNA processing of *MIR398* family genes is disrupted by their *cis*-NATs transcribed from coding genes.

Changes in the secondary or tertiary structure of RNA alter the stability of noncoding RNAs[23,49,50]. The suppression of *NAT398b/c* in the miRNA processing of *MIR398b/c* should be relevant to RNA secondary or tertiary structural deformation in the case of the existence of their NATs. These NATs may interrupt miRNA processing and undo the degradation or translational inhibition of their targets, in turn eventually influencing the key developmental processes of embryo, meristem, leaf and floral patterning[64,65]. Plants may exploit the balance between pri-miRNAs and their *cis*-NATs to facilitate plant development and stress response; any change in expression levels of *cis*-NATs can thus lead to an imbalance in biological responses. Understanding the complexity of the molecular relationship between pri-miRNAs and their *cis*-NATs is important. The existence and action of *cis*-NATs must be considered when *MIRNA* genes and artificial miRNAs are applied for the silencing of target genes in genetic transformation.

The expression of noncoding RNAs that transcribe in the opposite direction of protein-coding genes is often positively or negatively correlated with their cognate sense genes[31,52,53]. The cold-induced antisense transcript *COOLAIR* represses *FLOWERING LOCUS C* (*FLC*) transcription with increased H3K27me3 and decreased H3K36me3 levels in response to cold temperatures[52,66]. And the cold-induced antisense transcript *SVALKA-asCBF1* suppresses *CBF1* by RNAPII collision stemming[53]. The transcriptional activating mechanisms adopted by lncRNAs are varied. Previous research has shown that *MAS*, a NAT-lncRNA produced from *MADS AFFECTING FLOWERING4* (*MAF4*), recruits WDR5a and then guides COMPASS-like complexes to *MAF4* to enhance

histone-3 lysine-4 trimethylation (H3K4me3)[31]. Like *MAS*, *HOT-TIP* and *NeST* RNAs bind the adaptor protein WDR5 and recruit the MLL complex to maintain H3K4me3 and activate sense genes[67,68]. *LRK* antisense intergenic RNA binds histone modification proteins OsMOF and OsWDR5 to enhance H3K4me3 and H4K16ac in the *LRK1* gene region[69]. In addition, *EVX1as* increases the transcription of *EVX1* by facilitating the binding of the Mediator complex to the *EVX1* region, leading to an active chromatin state[70]. In this study, we found that noncoding genes *MIR398b* and *MIR398c* directly activate the transcription of their *cis*-NATs. Whether *MIR398b* and *MIR398c* regulate their cognate antisense genes through recruitment of COMPASS-like complexes or other mechanisms remains to be investigated.

In a previous investigation, overexpression of the *MIR398b* precursor sequence did not yield transgenic plants overexpressing miR398b, and the only recovered plants were those in which cosuppression had occurred[13]. In the present study, we overexpressed the pri-miR398b sequence and were likewise unable to obtain transgenic plants overexpressing miR398. Using the same primers to detect pri-miR398b levels, we found that pri-miR398b abundance was higher in our transgenic lines than in wild-type plants. This difference may be due to the overexpressed *MIR398b* sequence.

Heat stress is one of the major environmental stresses limiting plant growth, development and productivity; thus, plants have evolved special adaptive mechanisms to cope with high-temperature stress[71–74]. A previous study has shown that HSFA1b and HSFA7b bind directly to the promoter regions of *MIR398b* to activate the transcription of *MIR398b*[42]. In the present study, we found that heat stress activates the transcription of *MIR398b/c* genes and suppresses the transcription of *NAT398b/c* genes. Whether heat stress regulates *NAT398b/c* genes by recruiting heat stress transcription factors or via other mechanisms remains to be investigated. Our findings provide another example of a regulatory mechanism for plant thermotolerance, namely, through regulation of *cis*-NATs of *MIR398*. In particular, the upregulation of miR398 by heat stress is caused by stress-induced activation of *MIR398b/c* transcription and the downregulation of *NAT398b/c*.

miR398 is an important conserved miRNA that is proposed to be linked to various abiotic and biotic stresses[13,34,42–44]. Although this study has revealed the important role of *NAT398b/c* in regulating plant thermotolerance, the roles of *NAT398b/c* in regulating other abiotic and biotic stresses need to be investigated.

*NAT398b* is predicted to encode an acetylglucosaminyltransferase which suggests that NAT398b proteins have other biotic functions. *NAT398c* has additionally been reported to encode high-affinity nitrate transporter 2.7 (NRT2.7)[47], and NAT398c proteins play a specific role in nitrate content and proanthocyanidin accumulation in seeds[75,76]—which suggests that NAT398c proteins have other biotic functions. All of these results indicate that *NAT398b* and *NAT398c* are dual-functional. The regulatory relationship between *MIR398* genes and their *cis*-NATs is thus potentially useful to manipulate plant abiotic and biotic stresses tolerance. This fact adds an additional layer of complexity to the gene regulation of biological processes.

On the basis of our genetic and expression analyses, we propose a model for the regulatory loop between *MIR398* genes and their *cis*-NATs (Fig. 7). In this model, *MIR398b/c* genes and *NAT398b/c* are transcribed from opposing DNA strands at the same genomic locus, and *MIR398b/c* genes activate the transcription of their *cis*-NATs. *NAT398b/c* inhibit pri-miR398 processing by impairing the stability and accurate cleavage of pri-miR398b/c, thus causing abnormal miR398 processing and biological responses controlled by the miRNA-targeted genes.

Overexpression of *MIRNA* genes to silence miRNA-targeted genes is generally applied to enhance the yield and quality of

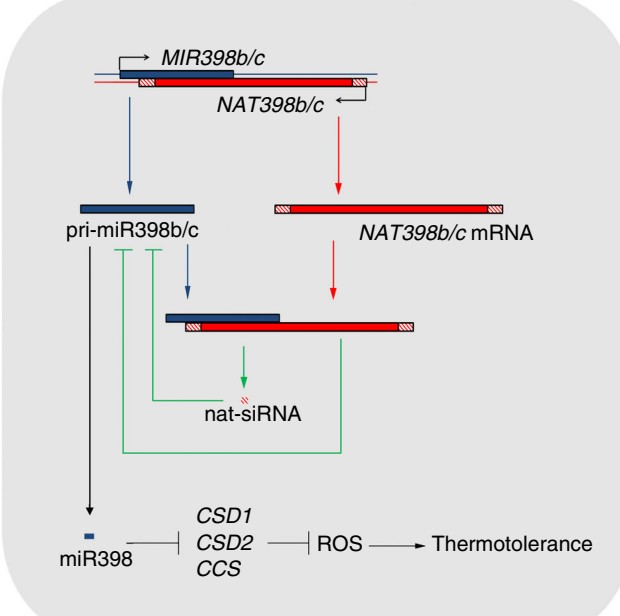

**Fig. 7 Proposed model for the regulatory loop between *MIR398b/c* genes and their *cis*-NATs.** *MIR398b/c* genes and their natural antisense transcripts *NAT398b/c* are transcribed from opposing DNA strands at the same genomic locus. *NAT398b/c* inhibit miR398 biogenesis and plant thermotolerance since they impair the stability and accurate cleavage of pri-miR398b/c.

agricultural crops[10]. The ability of *MIRNA* genes to suppress the processing of their miRNAs in the presence of their *cis*-NATs provides a reasonable explanation for the absence of accumulation of some miRNAs in transgenic plants overexpressing *MIRNA* genes. In addition, the effects of *cis*-NATs need to be avoided, or artificial miRNAs must be generated to silence their target genes. Because *MIR398/NAT398* genes are conserved in Brassicaceae, the optimized relationship between *MIR398* genes and their *cis*-NATs in Arabidopsis is potentially important for genetic manipulation of biological processes in various crops.

## Methods

**Plant materials and growth conditions.** *Arabidopsis thaliana* ecotype Columbia (Col), used as the wild-type, served as the genetic background for transgenic plants. SALK_033232 (*NAT398b-D*) and SALK_038698 (*mir398c*) were obtained from the ABRC stock center (http://signal.salk.edu/cgi-bin/tdnaexpress). Seeds were surface-sterilized and sown on plates containing Murashige and Skoog (MS) medium supplemented with 1% sugar. After 2 d at 4 °C, the plates were transferred to a phytotron under growth conditions of 22 °C and 16/8-h light/dark. For detection of heat response, 15-d-old seedlings were incubated at 38 °C, and plants grown at 22 °C served as controls.

**Plant sRNA sequencing.** For sRNA sequencing, 15-d-old seedlings of *A. thaliana* in MS medium were collected with two biological replicates. RNA samples were extracted using Trizol reagent (Invitrogen, Carlsbad, CA, USA) and treated with DNase I (Takara, Dalian, China) to remove DNA contamination. RNA samples prepared from seedlings were sent to WuXi AppTec (Shanghai, China) for sRNA sequencing.

**Gene cloning and generation of transgenic plants.** *MIR398a*, *MIR398b*, *MIR398c*, *NAT398b*, and *NAT398c* genes were amplified from Col genomic DNA, and the PCR fragments were subcloned into pCAMBIA1301 binary vectors along with a CaMV 35 S promoter and eGFP. Fragments corresponding to the promoters of *MIR398a*, *MIR398b*, *MIR398c*, *NAT398b*, and *NAT398c* were inserted into pCAMBIA1301 binary vectors upstream of β-glucuronidase (*GUS*) to replace the CaMV 35 S promoter. The destination vectors were introduced into *Agrobacterium tumefaciens* strain GV3101 (pMP90RK) using the freeze-thaw method and then transformed into wild-type (Col) plants. For selection of transgenic plants, seeds were sterilized and germinated on MS medium containing 25 mg/L hygromycin. Seedlings exhibiting resistance to hygromycin were transplanted and grown in a

greenhouse at 22 °C under a 16-h/8-h light/dark regime. Seeds from each transgenic plant were harvested separately for subsequent observation.

**CRISPR/Cas9 plasmid construction and mutant screening.** The 23-bp guide RNA sequences (including PAM) were selected within the target genes, and their targeting specificity was confirmed by a Blast search against the Arabidopsis genome (https://www.arabidopsis.org/index.jsp). The designed targeting sequences were synthesized and annealed to form oligo adaptors[77]. The vector psgR-Cas9-At was digested with BbsI and purified using a gel extraction kit. Ligation was carried out in 10-μL reaction volumes containing 10 ng of digested psgR-Cas9-At vector and 0.05 mM oligo adaptor, and the ligated product was directly transformed into *E. coli* competent cells to produce psgR-Cas9-At-1. The vector psgR-Cas9-At-1 was then digested with EcoRI and HindIII, and the validated psgR-Cas-At-1 fragment was inserted into a pCAMBIA1301 vector. GV3101 was selected for Arabidopsis genetic transformation.

Genomic DNA was extracted from T₂ transgenic lines, and the primers flanking the designated target site were used for PCR amplification. The PCR products were sequenced and Blasted to identify mutation sites.

**RNA extraction and transcript analysis.** Total RNA was extracted using Trizol reagent (Invitrogen) from seedlings in MS medium or rosette leaves grown in soil. cDNA was synthesized in 20-μL reaction volumes using 1 μg DNase-I-treated total RNA and oligo-dT primers or gene/miRNA-specific primers. Real-time PCR was performed in 20-μL volumes containing 3 μL of 10-fold diluted cDNA, 10 μL of iQ SYBR Green Supermix (Bio-Rad, Hercules, CA, USA), and 0.25 μM of each primer. The analysis was performed in a *MyiQ2* Two-color Real-time PCR Detection system (Bio-Rad) using the following cycling conditions: initial denaturation at 95 °C for 3 min, followed by 45 cycles of 95 °C for 10 s, 55 °C for 20 s, and 72 °C for 20 s. The primers used are listed in Supplementary Data 1. All data were normalized against expression levels of the *ACTIN2* gene or 18 S rRNA.

**Northern blotting.** Total RNA (16–30 μg) was separated on 19% polyacrylamide denaturing gels. The RNA was then transferred to a Hybond membrane (Amersham Biosciences, GE Healthcare) for 2 h at 200 mA. After crosslinking for 5 min with UV irradiation, the Hybond membrane was hybridized overnight at 42 °C with 3′ biotin-marked DNA probes complementary to the predicted miRNA or U6 sequences. Autoradiography of the membrane was performed using a Chemiluminescent Nucleic Acid Detection Module kit[78].

**GUS staining.** GUS staining was performed on 14-d-old plants. Seedlings of the transgenic plants were immersed in GUS staining solution (10 mM EDTA, 0.1% TritonX-100, 100 mM NaPO₄ [pH 7.0], 1 mM 5-bromo-4-chloro-3-indolyl glucuronide, 0.5 mM K₃[Fe(CN)₆], and 0.5 mM K₄[Fe(CN)₆]) and incubated at 37 °C overnight. Seedlings were then washed with 50% alcohol and fixed in water for further analysis[79].

**3,3′-Diaminobenzidine tetrahydrochloride (DAB) staining.** To detect H₂O₂ accumulation, 19-d-old soil-grown plants were subjected to 22 °C (control) or 38 °C (heat treatment) for 1 h. Detached leaves of plants sharing the same pot used for superoxide detection were vacuum-infiltrated with 1 mg/mL DAB in 1 M Tris-acetate buffer (pH 7.5). After incubating samples for 19 h at room temperature in darkness, chlorophyll was removed using 70% ethanol.

**Rapid amplification of cDNA ends (RACE).** Sequence information for *AtMIR398b/c* and their *cis*-NATs was retrieved from http://www.arabidopsis.org/. RNA samples were isolated from 3-week-old seedlings. After synthesis of cDNA using a SMARTer RACE 5′/3′ kit (Clontech), the 5′ and 3′ ends of pri-miR398b/c and their *cis*-NATs were amplified by RACE-PCR using gene-specific primers (Supplementary Data 1). The 3′ and 5′ PCR products were excised from the gel and cloned into a pMD18T vector (Takara). At least 20 positive colonies were sequenced for each RNA sample.

**Transient transformation assay.** Four-week-old tobacco leaves were inoculated with *Agrobacterium tumefaciens* GV3101 containing *MIR398a*, *MIR398b*, *MIR398c*, *NAT398b*, or *NAT398c* vectors. Three days later, RNA was extracted from the leaves for qRT-PCR.

**Ribonuclease protection assay.** RNA was treated with ribonuclease A + T at 37 °C for 30 min, followed by treatment with proteinase K at 37 °C for 30 min and then RNA phenol/chloroform extraction. A negative-control RNA was subjected to the same treatment. cDNA was synthesized and real-time PCR was carried out using different primers designed to target overlapping and nonoverlapping regions of sense and antisense transcripts.

**Analysis of *NAT398b/c* transcriptional activity.** Intron-specific and oligo-dT primers were used for reverse transcription following the method of Liu et al.[80]. We then applied intron-specific primers for detection of primary mRNA levels of

*NAT398b/c* genes in transgenic plants overexpressing *MIR398b* and *MIR398c* genes.

**Chromatin immunoprecipitation (ChIP) assays**. Seedlings of 3-week-old control and transgenic plants were fixed in 1% formaldehyde under vacuum. Fixed tissues were homogenized, and chromatin was isolated and sonicated. The solubilized chromatin was immunoprecipitated by adding Ser5P Pol II (ab5131, lot GR171392-7; Abcam, Cambridge, MA, USA) antibody for overnight incubation at 4 °C[74]. The amount of precipitated DNA was calculated relative to the total input of chromatin and expressed as a percentage of the total according to the following formula: % input = $2^{\Delta Ct} \times 100\%$, where $\Delta Ct = Ct$ (input) − Ct (IP), and Ct is the mean threshold cycle of the corresponding amplification reaction. The primers used are listed in Supplementary Data 1.

**RNA decay assay**. Two-week-old seedlings of Col and *p35S::NAT398b/c* were treated with 100 μg/mL actinomycin D (Sigma–Aldrich). Materials were harvested after 2 and 8 h. Total RNA was extracted and used for qRT-PCR assays.

**Thermotolerance assay**. To measure the survival rate of flowers under heat stress, 21-d-old soil-grown plants were subjected to 38 °C for 3 d, and the number of surviving flowers was recorded. To measure leaf thermotolerance, leaves detached from 16-d-old soil-grown plants were subjected to heat stress for 0 (control) or 7 h at 38 °C. The treated leaves were then photographed, and the number of green leaves was recorded 5 d later.

**Reporting summary**. Further information on research design is available in the Nature Research Reporting Summary linked to this article.

## Data availability

The data that support the findings of this study are available within the paper and its supplementary information. RNA-seq data associated with this study have been deposited in the NCBI SRA under accession PRJNA665283. Source data are provided with this paper. Any other supporting data are available from the corresponding author upon request.

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

## Acknowledgements

This work was supported by the National Programs for Science and Technology Development of China (Grant no. 2016YFD0101900) and the Natural Science Foundation of China (Grant nos. 31471883 and 31571261).

## Author contributions

Y.L. and J.Y. conceived the project; Y.L., J.Y., and Y.H. designed the experiments; Y.L. performed the experiments; Y.L., Y.H., and X.L. analyzed and interpreted the data; Y.L., J.Y., and Y.H. wrote the manuscript.

## Competing interests

The authors declare no competing interests.

## Additional information



