## [Peer Review File · Nature Communications]

Reviewers' comments:

Reviewer #1 (Remarks to the Author):

Li et al. explore the interplay between miRNA production and natural antisense transcripts (NATs) in plants. They identify heat-induced NATs to miR398 in brassicas that repress pri-miRNA398 processing. The authors propose a trans-acting effect of NATs on miRNA398 genes. Experimental modulation of NATs affects plant thermotolerance through miRNA target CSD genes. Moreover, the interplay between miRNA and NATs could form a regulatory loop. While non-protein-coding RNA (PCG) NATs have previously implicated in temperature dependent regulation of plant protein coding gene expression, we know less about the effect on miRNA expression. An interesting aspect of this paper is the suggestion of non-coding RNA-like NAT functions of a protein-coding gene, NAT398b. It may be argued that the effect of NATs on miRNA represents a specialized interest since PCG/NAT pairs vastly outnumber miRNA/NAT pairs. On the other hand, a strength of this paper is the suggested physiological relevance, albeit through previously described pathways acting through miRNA398/CSD.

Major points:

- the authors propose a trans-acting function of NATs on miRNA398, in part because attempted over-expression of mir398b/c constructs resulted in the surprising finding of reduced mir398b/c levels. To validate these findings, the authors should cross the 35S-mir398b lines with a mutant that does not generate NAT398b and repeat the analysis in these double mutants.
- The authors are missing a control in Figure 3 E/F. NA398b/c versions that contain mutations in the ORF to disrupt protein coding potential should be tested. These data will help to distinguish a NAT function from downstream effects of over-expressing a protein.
- The mechanistic data presented in Fig.4 are arguable the weakest part of the manuscript. More contemporary methods are available, for example data characterizing the changes in RNA structure. Based on the representations in Figure 1, it looks like the authors report dsRNA formation for the regions with sequence complementarity between NATs and precursor transcripts of mir398s. Since overlapping transcripts share reverse complement sequences, dsRNA formation is very likely outcome irrespective of whether this is functional or not. With some structural knowledge, specific point mutations could be devised that prevent binding and make this part much more convincing. Since the authors propose that this mechanism is trans-acting, the minimal additional control needed would be to generate a version of p35S-NAT that maintains the ORF, but loses the sequence complementarity to pre/pri-mir398. Perhaps this could be achieved by over-expressing a cDNA version of NAT398b.
- Line 337, the authors propose that mir398 directly activate the transcription of their cis-NATs. The authors suggest a highly controversial function for miRNA. What molecular mechanism are the authors proposing?

Minor points:

- In the abstract, line 60 Brassica instead of Braasica
- The potential contribution of nat-si-RNA needs to be clarified.
- The authors may wish to ask a native speaker to improve the English throughout the text.
- The authors should put their work into the context of plant temperature response regulation by NATs, especially during cold through COOLAIR at the FLC gene and SVALKKA at the CBF gene cluster (PMIDs: 31740502, 30385760).
- It seems like the NAT398b representation is inverted in 3b compared to 1b, and then back to the orientation in 1a in 4a. There are clear advantages of keeping one direction for readers and making the orientation more clear.
- Citations 14 and 15 are outdated, contemporary approaches to identify NATs more comprehensible should be factored in, particularly nascent transcription methods (PMID: 31863587).
- The images in Figure 6c/e should be presented so that an estimate of variation can be seen. These representations should include both transgenic lines presented in the previous figures, and a

measure that shows that heat treatment was equal between samples. For example HSP70 RNA levels.

Reviewer #2 (Remarks to the Author):

Review comments on

Li et al

Natural antisense transcripts of MIR398 genes suppress microR398 processing and attenuate plant thermotolerance

The authors of this manuscript presented a series of comprehensive analyses on the identification, expression, and transgenic manipulation of cis-natural antisense transcripts associated with MIR398 loci (cis-NAT398) and their possible regulatory role in Arabidopsis. Briefly, in an extended effort from a previous work done in Brassica rapa, the authors were able to identify over 20 cis-NATs of MIRNA genes in heat-induced B. rapa transcriptomic datasets. Through an in silico analysis, dozens of similar cis-NATs of MIRNA genes were also identified in Arabidopsis, of which cis-NATs of MIR398 genes were among a few that exhibited substantial evolutionary conservation beyond Brassicaceae. Next, it was shown that the locus-specific expression of cis-NAT398 appears to be attributable to the observed low miR398 expression levels in transgenic Arabidopsis in overexpression experiments. It was further shown that cis-NAT398 appears to suppress miR398 biogenesis through interference with pri-miR398 processing, most likely through formation of dsRNA. Curiously, overexpression of MIR398 appeared to activate cis-NAT398 transcription through an unknown mechanism. Finally, it was proposed that cis-NAT398 may play a regulatory role in miR398-mediated thermotolerance in plants.

Overall, the manuscript presented interesting and novel aspects of RNA-mediated regulatory mechanism for gene expression in plants. The experiments were mostly well thought out and rigorous in general. However, this reviewer believes that addressing the following questions would further improve the manuscript in making a more convincing case.

1. Although this manuscript focuses on cis-NATs associated with MIRNA loci (and on cis-NAT398 in particular), it is quite relevant to document and show if, and to what extent, the newly discovered cis-NATs were biased to MIRNA loci.
2. How would the authors reconcile the following seemingly obvious conflicting observations: (a) the cis-NATs were originally identified in heat stressed transcriptomes; and (2) miR398 expression was substantially induced upon heat treatment (Fig 6A)?
3. The authors did promoter analysis for both MIR398 and cis-NAT398 using GUS reporter. Did anyone or both of the reporter constructs exhibit heat-induced expression?
4. While data appeared to suggest that the cis-NAT398 likely function as long non-coding RNA (lncRNA) in regulating miR398 expression, it is unclear if these ORF-containing transcripts are also translated in vivo into predicted proteins (in which case these cis-NAT would be dual-functional). An in silico analysis on the existing proteomic data would help address this issue.
5. The mechanism by which MIR398 overexpression activates cis-NAT398 transcription is currently unknown and rather puzzling. Any hypothesis or speculated mechanisms would inspire future study.

Other minor points:

- a. In numerous cases, the wording in figure legends is overly redundant and needs to be condensed;

b. Based on the context, the word "detected" should be replaced by "examined" or "measured" in numerous cases.

Reviewers' comments:

Reviewer #1 (Remarks to the Author):

Li et al. explore the interplay between miRNA production and natural antisense transcripts (NATs) in plants. They identify heat-induced NATs to miR398 in brassicas that repress pri-miRNA398 processing. The authors propose a trans-acting effect of NATs on miRNA398 genes. Experimental modulation of NATs affects plant thermotolerance through miRNA target CSD genes. Moreover, the interplay between miRNA and NATs could form a regulatory loop. While non-protein-coding RNA (PCG) NATs have previously implicated in temperature dependent regulation of plant protein coding gene expression, we know less about the effect on miRNA expression. An interesting aspect of this paper is the suggestion of non-coding RNA-like NAT functions of a protein-coding gene, NAT398b. It may be argued that the effect of NATs on miRNA represents a specialized interest since PCG/NAT pairs vastly outnumber miRNA/NAT pairs. On the other hand, a strength of this paper is the suggested physiological relevance, albeit through previously described pathways acting through miRNA398/CSD.

Major points:

- the authors propose a trans-acting function of NATs on miRNA398, in part because attempted over-expression of mir398b/c constructs resulted in the surprising finding of reduced mir398b/c levels. To validate these findings, the authors should cross the 35S-mir398b lines with a mutant that does not generate NAT398b and repeat the analysis in these double mutants.

Answer: Good suggestion. *p35S::amiR-NAT398b* plant is equal to a mutant with defects in generation of NAT398b transcripts while *CRNAT398b* plant has no effect on the expression level of NAT398b transcripts. We crossed the *p35S::MIR398b* plants with *p35S::amiR-NAT398b* and *CRNAT398b* plants for analysis of genetic interaction between *MIR398b* and *NAT398b*. Expression levels of pri-miR398b and miR398 in *p35S::MIR398b* × *p35S::amiR-NAT398b* plants were significantly higher than those in *p35S::MIR398b* plants (Figure S6), whereas levels in *p35S::MIR398b* × *CRNAT398b* plants were not significantly higher than those in *p35S::MIR398b* plants (Figure S6). These results indicated that NAT398b transcripts rather than NAT398b proteins impaired the expression of pri-miR398b and miR398 in *p35S::MIR398b* plants. We have described this result in L229–L236.

- The authors are missing a control in Figure 3 E/F. NA398b/c versions that contain mutations in the ORF to disrupt protein coding potential should be tested. These data will help to distinguish a NAT function from downstream effects of over-expressing a protein.

Answer: Thanks for the suggestion. We added a control in Figure 3e/f.

-The mechanistic data presented in Fig.4 are arguable the weakest part of the manuscript. More contemporary methods are available, for example data characterizing the changes in RNA structure. Based on the representations in Figure 1, it looks like the authors report dsRNA formation for the regions with sequence complementarity between NATs and precursor transcripts of mir398s. Since overlapping transcripts share reverse complement sequences, dsRNA formation is very likely outcome irrespective of whether this is functional or not. With some structural

knowledge, specific point mutations could be devised that prevent binding and make this part much more convincing. Since the authors propose that this mechanism is trans-acting, the minimal additional control needed would be to generate a version of p35S-NAT that maintains the ORF, but loses the sequence complementarity to pre/pri-mir398. Perhaps this could be achieved by over-expressing a cDNA version of NAT398b.

Answer: Good suggestion too. *NAT398b* cDNA shares the reverse complement sequences in pre-miR398b while *NAT398b* CDS (coding sequence) fails to. To examine whether 3'-UTR regions of *NAT398b* affect the expression level of *MIR398b*, we overexpressed *NAT398b* cDNA and *NAT398b* CDS in Col. In this way, we generated a version of *NAT398b* that maintains the its ORF, but loses 315 nt sequence complementarity to pri-miR398 regions. In *p35S::NAT398b(cDNA)* plants, *NAT398b* was upregulated, concomitant with downregulation of pri-miR398b (Figure S10). In contrast, the upregulation of *NAT398b* in *p35S::NAT398b(CDS)* plants did not accompany downregulation of pri-miR398b (Figure S10). This result suggests that the *NAT398b* 3'-UTR region complementarity to pri-miR398b is crucial for pri-miR398b expression. We have described this result in L300–L309.

Line 337, the authors propose that mir398 directly activate the transcription of their cis-NATs. The authors suggest a highly controversial function for miRNA. What molecular mechanism are the authors proposing?

Answer: Here the function of *MIR398b/c* is achieved through pri-miR398b/c. pri-miR398b/c acts similar as lncRNA in regulating NAT398b/c. The molecular relationship between NAT and lncRNA has been studied and reported. We propose the pri-miRNA of non-coding genes *MIR398b* and *MIR398c* directly activate the transcription of their *cis*-NATs. Whether *MIR398b* and *MIR398c* regulate their cognate antisense genes through recruiting the COMPASS-like complexes remains to be investigated. We have discussed these in Discussion section L472–L485.

Minor points:

- In the abstract, line 60 Brassica instead of Braasica

Answer: “Braasica” is replaced with “Brassica”.

- The potential contribution of nat-si-RNA needs to be clarified.

Answer: I agree with you. nat-siR398b-1 and nat-siR398b-2 are located at the overlapped region of *NAT398b* with *MIR398b* (Figure 1b). Both nat-siRNAs accumulated at much higher levels in *p35S::NAT398b* plants than in the wild-type (Figure 4e and Figure S11b). The expression levels of both pri-miR398b and pri-miR398c declined in the *p35S::nat-siR398b-1* plants (Figure 4f). These nat-siRNAs regulate both miR398 biogenesis and *NAT398b* expression and thus may be affected at high and low temperature and act in plant response to abiotic stress. On the other hand, the constructs of nat-siR398b-1 and nat-siR398b-2 can be used to improve plant thermotolerance through genetic transformation.

- The authors may wish to ask a native speaker to improve the English throughout the text.

Answer: A English speaker polished English throughout the text.

- The authors should put their work into the context of plant temperature response regulation by NATs, especially during cold through COOLAIR at the FLC gene and SVALKA at the CBF gene cluster (PMIDs: 31740502, 30385760).

Answer: We cited these references in L359 and L468-L471 and discussed the difference between our result about *NAT398b* at *MIR398b* loci and *SVALKA* at the *CBF* gene cluster in Discussion section.

- It seems like the NAT398b representation is inverted in 3b compared to 1b, and then back to the orientation in 1a in 4a. There are clear advantages of keeping one direction for readers and making the orientation more clear.

Answer: NAT398b representation is readjusted as recommended.

- Citations 14 and 15 are outdated, contemporary approaches to identify NATs more comprehensible should be factored in, particularly nascent transcription methods (PMID: 31863587).

Answer: This reference was cited as hoped.

- The images in Figure 6c/e should be presented so that an estimate of variation can be seen. These representations should include both transgenic lines presented in the previous figures, and a measure that shows that heat treatment was equal between samples. For example HSP70 RNA levels.

Answer: We include both transgenic lines in Figure 6c/e. Heat treatment was equal between transgenic lines. HSP70 RNA levels were higher in heat-tolerant transgenic lines than in the wild-type.

Reviewer #2 (Remarks to the Author):

Review comments on

Li et al

Natural antisense transcripts of MIR398 genes suppress microR398 processing and attenuate plant thermotolerance

The authors of this manuscript presented a series of comprehensive analyses on the identification, expression, and transgenic manipulation of cis-natural antisense transcripts associated with MIR398 loci (cis-NAT398) and their possible regulatory role in Arabidopsis. Briefly, in an extended effort from a previous work done in Brassica rapa, the authors were able to identify over 20 cis-NATs of MIRNA genes in heat-induced B. rapa transcriptomic datasets. Through an in silico analysis, dozens of similar cis-NATs of MIRNA genes were also identified in Arabidopsis, of which cis-NATs of MIR398 genes were among a few that exhibited substantial evolutionary conservation beyond Brassicaceae. Next, it was shown that the locus-specific expression of cis-NAT398 appears to be attributable to the observed low miR398 expression levels in transgenic Arabidopsis in overexpression experiments. It was further shown that cis-NAT398 appears to suppress miR398 biogenesis through interference with pri-miR398 processing, most likely

through formation of dsRNA. Curiously, overexpression of MIR398 appeared to activate cis-NAT398 transcription through an unknown mechanism. Finally, it was proposed that cis-NAT398 may play a regulatory role in miR398-mediated thermotolerance in plants.

Overall, the manuscript presented interesting and novel aspects of RNA-mediated regulatory mechanism for gene expression in plants. The experiments were mostly well thought out and rigorous in general. However, this reviewer believes that addressing the following questions would further improve the manuscript in making a more convincing case.

1. Although this manuscript focuses on cis-NATs associated with MIRNA loci (and on cis-NAT398 in particular), it is quite relevant to document and show if, and to what extent, the newly discovered cis-NATs were biased to MIRNA loci.

Answer: In combination with RNA-seq data and annotated miRNA genes in *B. rapa* and *A. thaliana*, we carried out a large scale of alignment to search for DNA and RNA sequence pairs that are fully reverse complement to each other and transcribed from the opposite DNA strands. We used RACE PCR to define the ends of pri-miRNAs and cis-NATs and determined the longest pri-miR398b/c and NAT398b/c transcripts. Specifically, pri-miR398b and NAT398b transcript shares the full complementarity for 659 nt while pri-miR398c and NAT398c transcript shares the full complementarity for 1696 nt.

2. How would the authors reconcile the following seemingly obvious conflicting observations: (a) the cis-NATs were originally identified in heat stressed transcriptomes; and (2) miR398 expression was substantially induced upon heat treatment (Fig 6A)?

Answer: The cis-NAT398 was originally identified in heat stressed transcriptomes where it was downregulated at high temperature. Our RNA-seq, qRT-PCR (Fig 6b) and GUS signals analysis (Fig S13d) data showed that cis-NATs of *MIR398b/c* were heat-inhibited, and pri-miR398b/c were heat-induced. The upregulation of miR398 by heat stress is caused by stress-induced activation of *MIR398b/c* transcription and the downregulation of *NAT398b/c*. We discussed this in Discussion section L497–L506.

3. The authors did promoter analysis for both MIR398 and cis-NAT398 using GUS reporter. Did anyone or both of the reporter constructs exhibit heat-induced expression?

Answer: As suggested, we created the constructs of *MIR398b/c*- and *NAT398b/c*-promoter–GUS and bred the transgenic plants for analysis GUS activity under heat stress condition. Analysis of the seedlings revealed an increase in GUS signal intensity after stress treatment in *pMIR398b/c::GUS* but a decrease in *pNAT398b/c::GUS* (Figure S13d). These results indicate that heat stress activates the transcription of *MIR398b/c* genes and suppresses the transcription of *NAT398b/c* genes. We have described this result in L371–L377.

4. While data appeared to suggest that the cis-NAT398 likely function as long non-coding RNA (lncRNA) in regulating miR398 expression, it is unclear if these ORF-containing transcripts are also translated in vivo into predicted proteins (in which case these cis-NAT would be dual-functional). An in silico analysis on the existing proteomic data would help address this issue.

Answer: *NAT398c* encodes a high affinity nitrate transporter 2.7 (NRT2.7)⁴⁷, and *NAT398c* proteins play a specific role in nitrate content and proanthocyanidin accumulation in seeds⁷⁵,⁷⁶—which suggests that *NAT398c* proteins have other biotic functions. *NAT398b* gene is predicted to encode a 377- amino acid protein acetylglucosaminyltransferase but *NAT398b* protein has not been reported. We will check *NAT398b* protein. *NAT398b/c* would be dual-functional. We discussed this in Discussion section L511–L520.

5. The mechanism by which *MIR398* overexpression activates *cis-NAT398* transcription is currently unknown and rather puzzling. Any hypothesis or speculated mechanisms would inspire future study.

Answer: An interesting question. *Pri-miR398b/c* are likely to function as long non-coding RNA (lncRNA) in regulating *NAT398b/c* expression. Several studies showed that *MAS*, a NAT-lncRNA produced from the *MADS AFFECTING FLOWERING4 (MAF4)* recruits *WDR5a* and then guides the COMPASS-like complexes to *MAF4* to enhance histone 3 lysine 4 trimethylation (H3K4me3)³¹. *HOTTIP* and *NeST* RNA bind the adaptor protein *WDR5* and recruit the *MLL* complex to maintain H3K4me3 and activation of sense genes^{67, 68}. *LAIR (LRK Antisense Intergenic RNA)* binds histone modification proteins *OsMOF* and *OsWDR5* to enhance H3K4me3 and H4K16ac in *LRK1* gene region⁶⁹. And *EVX1as* increases the transcription of *EVX1* through facilitating the binding of Mediator complex to *EVX1* region, leading to an active chromatin state⁷⁰. *Pri-miR398* may directly activate their *cis-NAT398* transcription through one of similar pathway as mentioned above. We will define this question. We have discussed these in L472–L485.

Other minor points:

a. In numerous cases, the wording in figure legends is overly redundant and needs to be condensed;

Answer: Good suggestion. The figure legends were condensed.

b. Based on the context, the word “detected” should be replaced by “examined” or “measured” in numerous cases.

Answer: The word “detected” was replaced.

REVIEWERS' COMMENTS

Reviewer #1 (Remarks to the Author):

The authors have addressed my questions satisfactorily and strengthened their manuscript with additional experimental data.